# Emerging Scholars in Academia: An Analysis of the Impact of IAPSS *Politikon* in the Academic Careers of Its Authors

Ana Magdalena Figueroa [1,*] , Nzube Chukwuma [2] and Lumanyano Ngcayisa [3]

1   Legislative Assembly of the Republic of El Salvador, La Libertad 01511, El Salvador
2   School of Politics and International Studies, Central China Normal University, Wuhan 430079, China
3   Department of Political Studies & Governance, University of the Free State, Bloemfontein 9301, South Africa
*   Correspondence: ana.figueroa@asamblea.gob.sv

**Abstract:** This paper analyzes students, recent graduates, and emerging scholars' involvement in academic publishing, specifically by studying the impact of their publications in *Politikon*, the International Association for Political Science Students' Journal, and their academic careers. The results of a standardized self-administered survey serve to assess authors' motivations and impact of publish ing with IAPSS *Politikon*. The data show that publishing with *Politikon* has helped emerging researchers get more experience concerning the publication process and has improved their research, writing, and analyzing skills. Additionally, an essential part of them said they increased their educational background or obtained a new higher position after publishing with *Politikon*. In fact, 50 per cent of these scholars said they were promoted after publishing with the journal. Furthermore, *Politikon*, establishes a platform for Global South scholarship to be at the forefront of Political Science/International Relations debates and knowledge production. This implies the importance of *Politikon* in the early career of scholars by giving them the right tools to develop professionally and by reaching scholars from all around the world, especially from the Global South in an effort to contribute to global international relations and global governance reform.

**Keywords:** emerging scholars; IAPSS *Politikon* journal; recent graduates; Global South



## 1. Introduction

Academia is a competitive environment where emerging scholars feel pressured to "publish or perish". These scholars comprise university students such as bachelor's, master, and PhD students. Publishing in academic journals includes original research and review articles, clinical case studies, clinical trials, perspectives, opinions, commentary, and book reviews [1]. Emerging scholars may publish in open access or subscription-only journals, the former having higher readability by promising easy and rapid publication in return for authors to pay low article processing charges (APC), while the latter is where individual readers or institutions subscribe to articles after publication. Undoubtedly, there is a misleading impression of open access journals exploited by predatory journals [2]. The assumption, unfortunately, taints the image of all open access journals, including respectable open access journals indexed in the Scopus journal index [3], and implies that they are somehow inherently predatory [2]. There is an increase in the number of journals that are being published by fewer publishers, and because of this, the authors infer that the problem of predatory publishing is getting worse, and that this could be mostly due to mega-predatory publishers that have increased the number of journals they publish [4].

Despite this assumption on open access journals, Reference [2] sees predatory journals as deceptive journals that take the appearance of legitimate peer-reviewed journals and impact academic stakeholders by exploiting the open access model while using misleading strategies to solicit article submissions. The shock is that these fake practices by predatory journals as "scammers" that publish articles or unscientific papers for financial benefits [5] have increased sporadically from 53,000 in 2010 to 420,000 articles in 2014, with the current

estimated list of over 16,000 journals and an annual increase of 1800 [2]. In recent years, open access journals have had different classifications, such as Green open access, Gold open access, Diamond open access, Hybrid open access, Bronze open access, Black open access, and most times, these types of available accesses offer authors the opportunities to receive more citations depending on the level of the open access journal.

Therefore, part of the pressure of publish-or-perish culture comes from the changing academic landscape that links the increased number of publications to the academic promotion and other financial and personal benefits for authors. The type of journal, open access or subscription-based, factors such as peer-review-related charges, higher visibility of research papers, journals' prestige and citations, and speed from date of submission to acceptance and decision, are factors for students, recent graduates, and emerging scholars to consider before submission of manuscripts. There is also another important factor for young scholars to choose a journal to publish in: the information they gather from their environment due to the lack of resources. Consequently, the academic circle's factored pressure has led many academics to choose publication in questionable journals in a self-organized manner instead of pursuing a reputable alternative.

It is important to mention that this paper will use the term "emerging scholars" to describe those scholars who have recently graduated and are beginning their research careers. However, this paper will also explore other concepts in the literature. In recognizing this pending challenge for academia, *Politikon*, the International Association for Political Science Students' flagship publication, emerged. Politikon, as an alternative over the years, is being managed by emerging scholars from different areas of political science and international relations with the support of an international Advisory Committee of Senior Scholars. The emerging scholars contribute as reviewers, assistant editors, and editorial board members. Since its emergence in 2001, *Politikon*, a leading academic peer-reviewed journal, has had 54 volumes of scholarly articles published four times a year and distributed online free.

This paper seeks to analyze students, recent graduates, and emerging scholars' involvement in academic publishing, specifically by studying the impact of publication in *Politikon* on their academic careers. It will present the results of a general survey distributed among the authors who have published one or more articles with *Politikon*. Some of these authors have worked on the editorial team of *Politikon*, being themselves emerging scholars, while some of the authors have never had a relation to the journal.

This career is a brutal expectation of academic affairs and the profession's various gatekeepers [6] who face the intense pressure that, in order to progress in all ramifications, they must "publish or perish" [7] to enhance the strategic career development for society members [8]. The pressure and culture of publishing or perishing to succeed in academia [9] have severe consequences for academic research integrity [10] based on publishing that substitutes the "taste for science" with a "taste for publication", as citations dominate research rankings [9]. Yet researching and publishing have served as conditions for employment, promotion, and even maintaining one's job [11], as the publishing choice of emerging scholars may produce a dramatic maximized career outcome and increase career success [12].

For researchers, notably emerging scholars, the impact of researching and publishing is at the heart of career prospects to attain career goals [13] and to sustain a position in a career [14] that has been driven partly by institutional expectations and personal career ambitions [15]. Emerging scholars are individuals pursuing academic research at the sub-tenure level, regardless of years of experience [16]. These individuals can be categorized as students, recent graduates, and emerging scholars within the bachelor's degree, master's degree, PhD, and post-doctorate levels. Thus, we define emerging scholars as academic apprentices in the field of academia to build their careers and contribute meaningfully to societal development.

Across countries and universities worldwide, scholars have identified the im- portance of the attitude and practices of emerging scholars [17] and their growing number as the

biggest group of researchers [18]. Thus, while emerging scholars make up a large proportion of the academic workforce, and their experiences often reflect the broader culture of the research community [19], the consensus is that emerging scholars try to go beyond the first fragile rungs of the academic career ladder [20]. Hence, researching and publishing have remained problematic and complex for emerging scholars because of the problem of research funding [21] and the deficiencies of research and writing skills [22]. Other problems include language barriers to meeting a competitive career, pressure to network and career uncertainty [23], publishing in subscription journals [24] or open access journals [25] that published unscientific papers [5] with flagrantly questionable research practices [26].

How these problems shape and affect emerging scholars' career development has reinforced journal platforms' decision to increase emerging scholars' chances for publication and advance emerging scholars' career development and provide opportunities for contributing to scholarly debate. Therefore, significant efforts by stakeholders to catalyze systematic change in research culture and practice on emerging scholars [27] have taken place through scientific journal training invitations on co-reviewing and ghostwriting manuscripts [28]. This is done in efforts to tap into emerging scholars' potential as reviewers in the future and an opportunity to develop and refine emerging scholars' writing skills to enhance career and other professional development [9].

Given that, top journals and emerging journals, including universities journals, have created an initiative to guide emerging scholars' career research sufficiently. To this end, Palgrave Macmillan [29] offers emerging scholars on-campus opportunities relating to the publishing process and developing an academic career, and the Springer Nature [30] avenues to maximize and prioritize emerging scholars' academic knowledge and societal impact [31]. Other publishing platforms [32,33] have supported emerging scholars' career engagement in society to strengthen the voices of emerging scholars and fill the gaps in career development (Genetics Society of America, n.d [34]). They have contributed to emerging scholars' career productivity and skills growth through these opportunities [35].

Despite this journal's contribution to emerging scholars and scholarly discussion of emerging scholars, little is known about how the emerging scholars' research and publications influenced their careers in emerging scholar-led journals. However, research has investigated how emerging scholars' writing engagements in top journals and incremental and valuable contributions to the research spectrum impact societies despite emerging scholars' uncertainty in research outcomes [19]. However, research is limited on journal-driven emerging scholar initiatives that have affected their career. Given this, *Politikon,* the International Association for Political Science Students' flagship publication, surfaced to recognize emerging scholars' driven research and quest for career development amidst pending challenges. Since its emergence in 2001, *Politikon,* a leading academic peer-reviewed journal, has had 54 volumes of scholarly open access articles published four times a year. Consequently, over the years, Politikon has been managed by emerging scholars from different areas of political science and international relations with the support of the International Advisory Committee of Senior Scholars.

The understanding is that while emerging scholars are key drivers and contributors in Politikon and serve as reviewers, assistant editors, and editorial board members, the significant impact of these engagements towards career development remains understudied. This paper's overall objective analyzes emerging scholars' involvement in academic publishing by studying the impact of publication in *Politikon* on their academic careers. This paper will present the results of a general survey distributed among the authors who have published one or more articles with *Politikon.* Some of these authors have worked on the editorial team of *Politikon,* they themselves being emerging scholars, while some of the authors have never had any relation to the journal. Therefore, the remainder of the study is in five sections. The first will conceptualize emerging scholars in *Politikon.* Next, the study will review emerging scholars research engagements, and then third, the study methodology. Following this is a discussion of the results, analysis findings, and finally, the conclusion.

## 2. Literature Review

Students, recent graduates, and emerging scholars are recognized for being amongst the most active and creative researchers, due in part to the imposed need of the market to publish or perish, and they contribute in this way to an enormous pool of worldwide capacity that plays a central role in the knowledge of the [36] (pp. 99–111). Dissemination of knowledge is paramount to the advancement of science—journals are, therefore, an essential factor in academia, making that knowledge available to the global scholar community. Emerging researchers—students, postgraduates, and emerging scholars—are a part of that community and, as such, are just as interested in diffusing their work.

First, it is necessary to clarify what it means to be classified as a graduate and emerging scholar. Graduates have skills and qualities instilled by their university during their degree, including technical knowledge, soft skills, and disciplinary expertise [37]. In turn, graduates are agents of social good and contributors to social understanding. The Global Young Academy defines "emerging scientists" as those who have already acquired a PhD and have been pursuing their research career for up to 10 years, placing their age between 30 to 40 years [38], while [18] (p. 20) emerging scholars are "[r]esearchers who are generally not older than 35, who either have received their doctorate and are currently in a research position or have been in research positions but are currently doing a doctorate. In neither case are they researchers in established or tenured positions. In the case of academics, they are non-faculty research employees of the university".

Other publications [39] rely on age to define what constitutes an early career scholar (30 years old), and the EU is concerned with stage of education, defining emerging scholars as those in the first four years of their full-time research careers who have not obtained a PhD. For this study, this is the group we are focused on, as that is also the group invited to submit to *Politikon*.

This community of scholars is seen as creative, innovative, energetic, and mobile, constituting a global talent pool [38] (p. 7), technologically savvy and IT proficient, and web 2.0 users [40]. It is also growing in numbers and geographical scope. Nonetheless, there is a lack of studies focusing on emerging scholars overall, and not very many are concerned with their publication or career advancement strategies. Most studies to date were also biased towards those located in Europe and North America [38], a shortcoming that some scholarship has attempted to tackle [41].

Over the centuries, research has exclusively been centered around Global North scholarship. Seemingly, few debates disagree that the world today has inherent ramifications from colonialism [42] (pp. 2–3). Since globalization and tech nological advances have improved research output and scholarly collaboration, IAPSS is at the precipice of providing a platform for Global South scholarship.

Noticeably, Global South scholarship is becoming more concerned with their understanding and interpretation of non-Western schools of thought in contextualizing aca- demic knowledge; this is known and discussed as global international relations [42] (pp. 2–3). Notably, Western scholarship has significantly defined indigenous knowledge of non-Western societies while describing it using foreign languages to today's Global South community. These explanations and interpretations combined with Western theories have explained that the "Third World" was once what we know today as the Global South. Thus, Global IR [43] enables pluralistic universal ism, meaning it opens channels of vast knowledge systems across the globe while cautioning against a blanket approach and recognizing diversity. Arguably, IAPSS is on the precipice of being a catalyst for empowering non-Western scholarship, which can help change this narrative by contributing to the study of Political Science and International Relations from across diverse scholarships and perspectives.

Nevertheless, increased mobility is also associated with increased global competition. Research priorities have moved to the transnational level [38], with amplified international collaboration and heightened competition for available positions, funding, and access to data. Emerging scholars also must navigate different higher education systems across

nations, with other practices and demands, constantly adapting to local, national, regional, and global circumstances. A pressing result of this competition is precarity, with young researchers facing the slight possibility of full-time employment, but instead having to engage with a variety of part-time, contracted, non-tenure-track positions based on short-term funding acquired through a variety of national and international sources. Young researchers [40] are the most vulnerable population in the scholarly community. In this context, publishing has emerged as a powerful mechanism to cope with these challenges, as a means of building reputation, acquiring funding, accessing scholarships or employment, ensuring mobility, and enhancing the possibility of career progression.

In contrast, Global South scholars are underrepresented in international peer-reviewed scientific journals [44]. In comparison, Global North scholars have a greater outreach than Global South scholars, ostensibly bringing them to the forefront of the production and dissemination of knowledge. As a result, Global South scholars are marginalized from academic discussions and debates in their fields. Seemingly, the positive impact of IAPSS *Politikon* can be a channel to change this narrative, enabling a platform for global IR and wider/broader sources of knowledge systems to the study and activities of politics and international relations. Subsequently, widening the sources of knowledge systems better equips scientists, politicians, and policy-makers of the solutions available across the globe. Moreover, this allows for Global South members to look at non-Western/Eurocentric solutions.

Moreover, in academic publishing, the Oldenburg publishing model is still the norm, with academic journals fulfilling four core functions: introduction of new research, its dissemination, quality control (through peer-review), and archival record [45]. This model guarantees trustworthiness, authority quality, and dimensions irreplaceable for the academic community [39]. It has been argued that academics remain fixated on article publication, with universities actively promoting this practice, even if this is being increasingly contested [46]. Unsurprisingly, publishing remains of the utmost importance to young researchers—even through a growing awareness of the unbalance of the current publishing system [41].

The number of publications is the number one criterion for emerging scholars career advancement [38], followed by the journal's reputation for publishing their papers. These aspects were more valorized than making a scientific breakthrough. Peer review emerges as a guarantor of journals' reputation and a marker of trustworthiness, with emerging scholars seeking privileged peer-review journals when deciding on where to submit a paper. Other factors were also considered as necessary by this group of scholars, mainly whether or not the journal was open access (OA)– with emerging scholars, unlike their senior colleagues, privileging OA–, number of citations—or impact factor—and country where the journal was based, with emerging scholars choosing to publish in journals based in countries known for the quality of their research [39].

The advantages and disadvantages of peer review, as perceived by scholars, with the advantages being trustworthiness, quality feedback leading to paper improvement, and organization by experienced publishers [47]. On the other hand, the disadvantages are that they are slow and sometimes intimidating. There is also the possibility of "hands-off editors", unwilling or unable to be the ultimate judge of the quality of the paper and selecting inadequate or biased reviewers. The inconsistent quality of peer review allows for the publication of bad articles, legitimized by the fact that they went through the peer-review process—a concern mainly expressed by junior scholars.

Young researchers then believed more strongly that the peer-review process had become less rigorous. They were also less skeptical of OA, results that were complemented by a posterior study [18], which found that, when emerging scholars cannot publish in the top journals in their fields, with the impact factor emerging as particularly significant in their decision to submit a paper, they deploy other selection strategies—they are likely to submit their articles to journals where their chances of publication are higher, and journals that they have had a previous positive experience, with quicker response times and extensive feedback, appropriate audiences, and open access. It is primarily young and inexperienced researchers from developing countries (India, Nigeria, Turkey, followed

by some African and Middle Eastern countries) who publish in this type of outlet, which the authors at tribute to structural conditions: economic inequalities and sociocultural practices, including pressure to publish by academic institutions, with minimal regard for the quality of the publishing outlet [48,49]. These articles are often then included in university databases following the pub lication of an article by an author associated with the institution, thereby sanctioning and institutionalizing the practice. Global South scholarship requires significant support to expand its outreach. The question then is whether IAPSS, through its political science journal, *Politikon*, does just that. Ostensibly, IAPSS *Politikon* can be a platform to change the status quo if it provides a platform for a wider and broader outreach of sources of knowledge to political science and international relations.

Therefore, it is not surprising that, despite considerable differences in funding for R&D activities, the proportion of scientific papers by authors from developing countries published in established bibliometric databases has increased significantly [38]. Nonetheless, authors from this part of the world still search for legitimate publication opportunities. Politikon aims to present itself as one such option. The investigation, therefore, seeks to answer the following research question: What is the impact of Politikon and IAPSS journal publication on the academic careers of stu dents, recent graduates, and emerging scholars?

## 3. Methodology

It is important to begin this section by stating that the purpose of this paper is to assess the impact of publishing with *Politikon* in the careers of the emerging scholars who have successfully concluded the process with the journal, and this is why we are considering this journal as a case study. Therefore, being a case study, we consider that this paper has the quantity of answers required. A focused, self-administered survey was designed to provide an insight into authors' educational and professional experiences with the journal. Research has shown that this type of survey can motivate the participants to report issues they would not say or misreport in an interview [50]. The web survey also allowed us to reach more potential respondents, particularly the relevant geographical distributions of given respondents. Other advantages of this type of survey include the tendency to be more enjoyable, especially to young respondents [50], and to provide better and more comprehensive recommendations.

The standardized web survey was designed through the SurveyMonkey platform. It contains 27 questions on the impact and the experience of publication with *Politikon*, including the assistance provided by the editorial board and some demographic questions [51]. The survey was conducted from 10 June to 31 December 2021, and collected 33 responses. It is worth noting that the data were collected in two waves, the first wave being from 10 June to 18 June, and the second wave being from 1 December to 31 December. All respondents confirmed that they have published at least one article with *Politikon*. This survey uses volunteer sampling [52], as the authors receive the link but it is not mandatory for them to successfully finish the survey. As a means of collecting high-quality data, the survey included a variety of questions, such as open-ended questions, close-ended questions, ranking, etc., and respondents were given the opportunity to provide further information in cases where their answer is not contemplated in the design of a given survey question [53].

Furthermore, to avoid common problems of nonresponse or misreporting, the web survey was designed to provide privacy, anonymity, and confidentiality [50]. That is why demographic questions (age, nationality, gender, education level and field of expertise or significance, English proficiency, employment status, and income range) were placed at the end of the survey. The provision of contact information was deemed voluntary, as were the demographic questions asked to respondents.

The survey intended to uncover, firstly, emerging scholars' knowledge on *Politikon*, how authors got to know about the journal, and the reasons that motivated them to publish with it. Secondly, the survey sought to discover how publishing with *Politikon* helped these emerging scholars in their journey through academia, especially those in the Global South, given the fact that most of the journals in this topic tend to focus on Western scholars.

Thirdly, it contained questions about computer literacy and the authors' computer use habits in order to establish how inclusive the journal is concerning what the authors publish. Other questions inquired about the phases of the publication process itself and how helpful it was for the authors, particularly in terms of contact with the editors and editorial board members, and the advice received from the peer reviewers and the editorial team. The survey also included a question on authors' involvement with IAPSS to assess the extent to which respondents became more involved with the journal and its publisher as part of the reviewers' team, the editorial team, or any other capacity.

## 4. Results

It is important to remember that our purpose is to assess what has been the impact of publishing with *Politikon* in the careers of the emerging scholars, and this is why we are using this journal as a case study. Figure 1 shows the author demography: 54% of respondents were male, 36% were female, and less than 10% preferred not to disclose their gender. These results might correspond to the fact that there are more men in political science. However, there is still a great amount of work to be done to close the gender gap in academia and in the field of political science and international relations. With respect to the age of the respondents, 3% of the authors are age is 20-years-old or younger, thus possibly corresponding to bachelor's degree students. Most of the authors range from 20 to 29-years-old, while 42% of them are between 30 and 39-years-old.

| Gender | Frequency | Percentage% |
|---|---|---|
| Male | 18 | 54.55 |
| Female | 12 | 36.36 |
| Prefer not to disclose | 3 | 9.09 |
| **Total** | **33** | **100** |

| Age | | |
|---|---|---|
| Less than 20 | 1 | 3.03 |
| Between 20 and 29 | 16 | 48.48 |
| Between 30 and 39 | 14 | 42.42 |
| Between 40 and 49 | 2 | 6.06 |
| **Total** | **33** | **100** |

| Educational Status | | |
|---|---|---|
| High school | 1 | 3.03 |
| Bachelors | 4 | 12.12 |
| Masters | 16 | 48.48 |
| PhD | 12 | 36.36 |
| **Total** | **33** | **100** |

| Educational Majors | | |
|---|---|---|
| Political Science | 21 | 63.64 |
| International Relations | 9 | 27.27 |
| Economics | 2 | 6.06 |
| Law or International Law | 1 | 3.03 |
| **Total** | **33** | **100** |

**Figure 1.** Demography of Respondents. The questions of this survey can be found in the Supplementary Materials. Source: Prepared by the authors with data obtained from a survey (2021).

It is interesting to see that 48% of the authors are master's degree students, 36% are PhD candidates, and 12% are br's degree students. It is interesting because it allows us to see what stage of their career these emerging scholars are in, and this enforces the fact that most of them begin to publish while studying for their master's degree or PhD. One rather expected piece of data is that 64% of the authors majored in political science, and 27% of them majored in international relations, which are the focus subjects of the journal. On the other hand, 6% majored in economics, and 3% of them majored in law or international law. In other words, the data indicate that the survey respondents were relatively diverse in terms of gender, age, and educational status. However, most of them held a graduate

degree or a PhD, and studied the core discipline of the journal, political science. Still, it is worth noting that the journal also captures the experiences of undergraduate students whose works were published in the journal or of those who studied disciplines other than political science/international relations.

Regarding their nationality, the authors come from 20 different countries, notably four continents of the world: Europe with 42.4%, the Americas with 27.3%, Africa with 18.2%, and Asia with 12.1%. The indication is that many scholars are contributing from different countries publishing with *Politikon* (see Figure 2). What is particularly interesting here is that if we analyze the countries of the Global South that these authors are from, they account for 42.42% of articles published. This means that very close to half of the authors that have published in Politikon belong to countries in the Global South, showing that this journal seeks to reach those who are in countries other than those that are "normally" publish. For example, when we factor in countries in Africa, Asia, and some countries in the Americas, this implies a shift in the notion of Eurocentric and Western-centered scholarship.

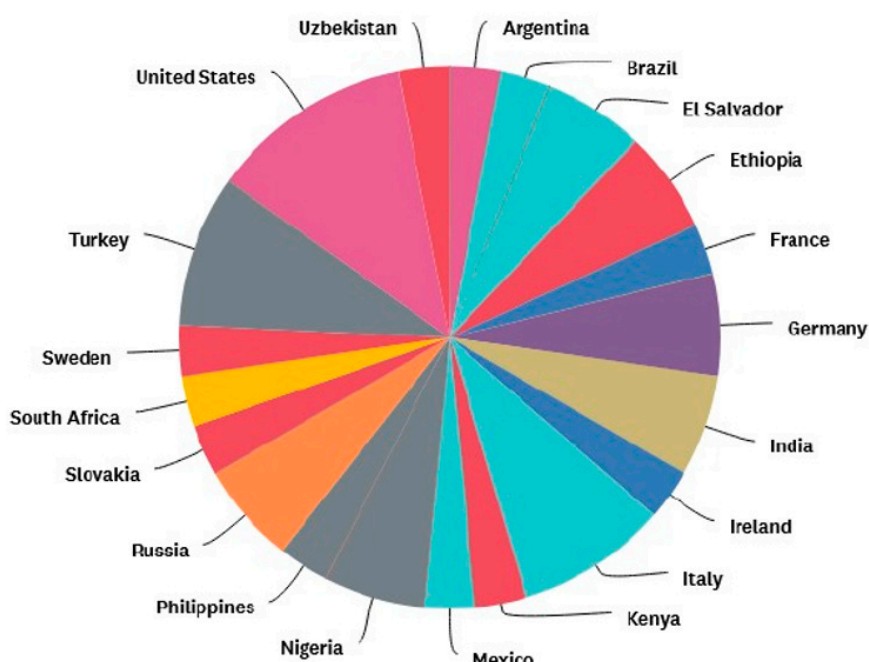

**Figure 2.** Countries of respondents. Source: Prepared by the authors with data obtained from a survey (2021).

Figure 2 (countries of respondents) illustrates the various countries of the participants of the study. The respondents' pool is vast, ranging from Western, Global North, to the Global South. The following countries have 3% representation, respectively on the chart: Argentina, Brazil, France, Ireland, Kenya, Mexico, the Philippines, Slovakia, South Africa, Sweden, and Uzbekistan. On the other hand, the following countries have 6% representation, respectively: El Salvador, Ethiopia, Germany, India, Nigeria, and Russia. Further, Italy and Turkey comprised 9%, respectively, and lastly, the United States as 12%. The wide range of participants depicts the inclusivity of the IAPSS *Politikon* journal. Although the highest percentage of participants in one country are from a Western country, the combination of all Global South countries including those in Latin America, Africa, and Eastern Europe arguably illustrates that Global South scholarship is empowered by IAPSS while prioritizing more global perspectives.

In Figure 3, respondents were asked how they heard about *Politikon*; 30% of stated that they learned of *Politikon* through the Google search engine while 47% said they heard of *Politikon* through the IAPSS website. About 6% discovered *Politikon* through the International Political Science Association (IPSA), and 11% learned through other sources. Moreso,

the survey showed that less than 3% knew about *Politikon* through social media and employer/university mediums of information. The impression is that while *Politikon* gains more attention through the IAPSS activities, there could also be more general promotion, as 84% of the respondents agreed to recommend their friends and col leagues to publish with *Politikon* (see question 17 in Supplementary Materials).

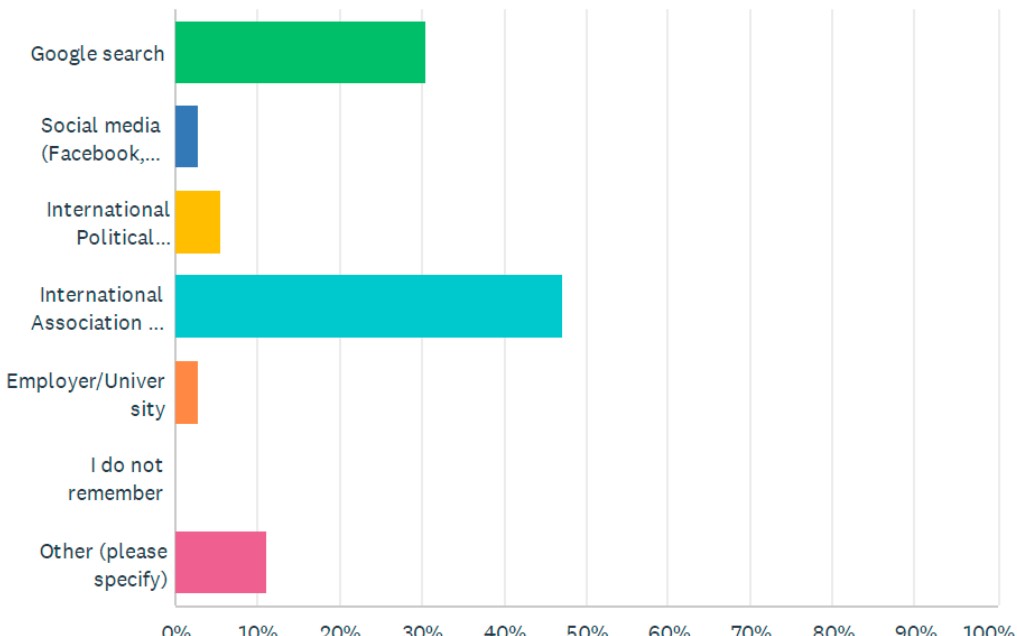

**Figure 3.** Q1. How did you hear about IAPSS *Politikon.* Source: Prepared by the authors with data obtained from a survey (2021).

In Figure 4, the reason for publishing with IAPSS *Politikon* was put forward to respondents. About 42% of them believed that the prestige and IAPSS journal's high-quality result and impact guided their choice of journal. Some respondents, about 39%, identified the prestige of the student's association (IAPSS) itself. Additionally, 19% saw the choice for open access as a determinant, while 8% were recommended to publish in *Politikon* and 13% for other unspecified reasons. Besides this, about 57% of our respondents were very much satisfied with the frequency of *Politikon* correspondence from the editorial board members (see question 18 in Supplementary Materials).

While *Politikon*'s overall prestige and correspondences were critical, Figure 4 shows which indexing services are essential for the authors when submitting to a journal. It was found that 50% of the respondents said that Scopus is vital to them, followed by the Web of Science ESCI with 27%. ProQuest followed with 9%. It is worth saying that *Politikon* is indexed in the Directory of Open Access Journals (DOAJ), ERIH PLUS, International Political Science Abstracts, and J-Gate (see Figure 5). A way of interpreting this is that it seems that even those to whom certain indexing services are essential sometimes choose journals that are not indexed, indicating that other reasons affect the choice of journal as well. In other words, from Figure 5, it is noticeable that authors/participants of this study are familiar with indexing services and can differentiate between indexed and non-indexed journals (these range from peer-review to predatory journals). Furthermore, at times, the prestige of the journal its perception of being welcoming, its inclusive platform, or its promise to obtain quality feedback influence a prospective author's choice.

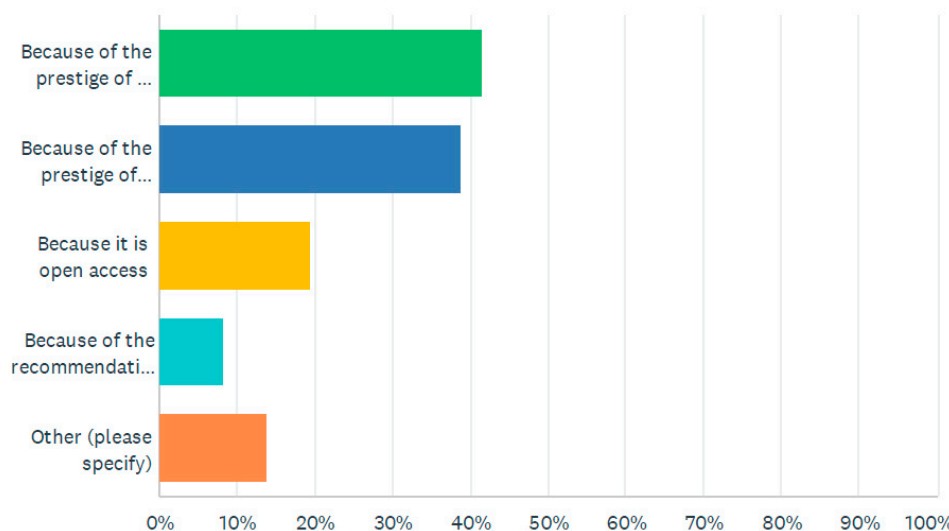

**Figure 4.** Q4. Why did you choose to publish with IAPSS *Politikon*. Source: Prepared by the authors with data obtained from a survey (2021).

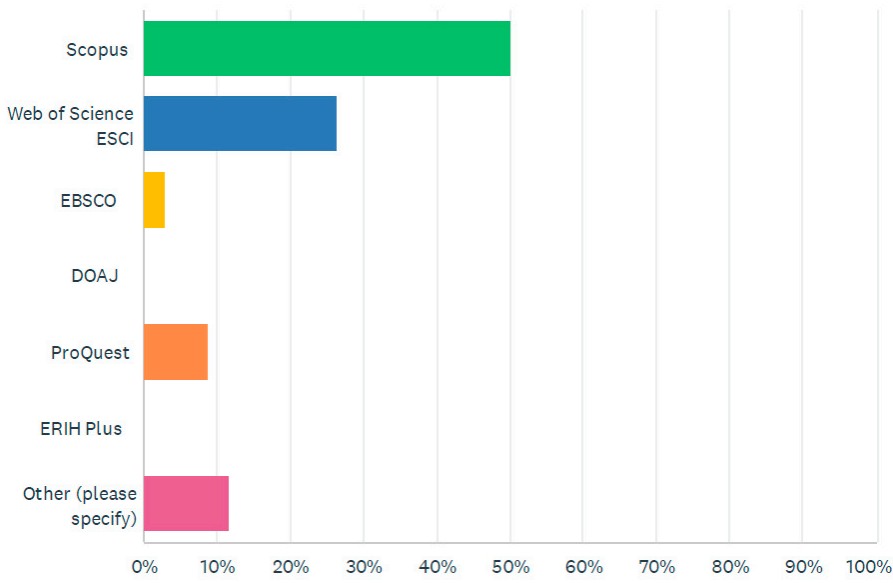

**Figure 5.** Q15. Which indexing services are essential for you to submit to a journal. Source: Prepared by the authors with data obtained from a survey (2021).

Furthermore, 87% of respondents reveal their preferences for journals with broad dissemination of research papers online once published. In such a manner, respondents were further asked about the nature of the corrections required for the manuscript submitted, and 42% indicated that they required minor corrections and 14% indicated that they required major corrections (see questions in Supplementary Materials). Perhaps the editorial process in *Politikon* was responsible for why respondents indicated that they would publish with *Politikon* again in the future.

Figure 6 shows the rating of the components or phases of the publication process with the journal and how helpful that part was for them. The rating goes from 1 to 5, where 1 corresponds to "strongly disagree", and 5 corresponds to "strongly agree". The results indicate high satisfaction with the publication process, especially with the first contact with the Editor in Chief, as well as with the Editorial Board Member, and the peer-review process scoring slightly better than the subsequent editing process and the clarity of the publication guide lines. In general, their satisfaction was high, hence 58% said

that they would recommend publishing with *Politikon*, while 37% said they might do so. Furthermore, 85% said they would recommend their friends or colleagues to publish with the journal. After publishing with the journal, 88% of authors disseminated their article, and most of them did this through Twitter, Facebook, and Research Gate. This indicates that the authors were proud of their publication and keen to share it with their networks. Additionally, an advantage of sharing their articles on social media is that social media and the open access nature of the journal increase the author's visibility.

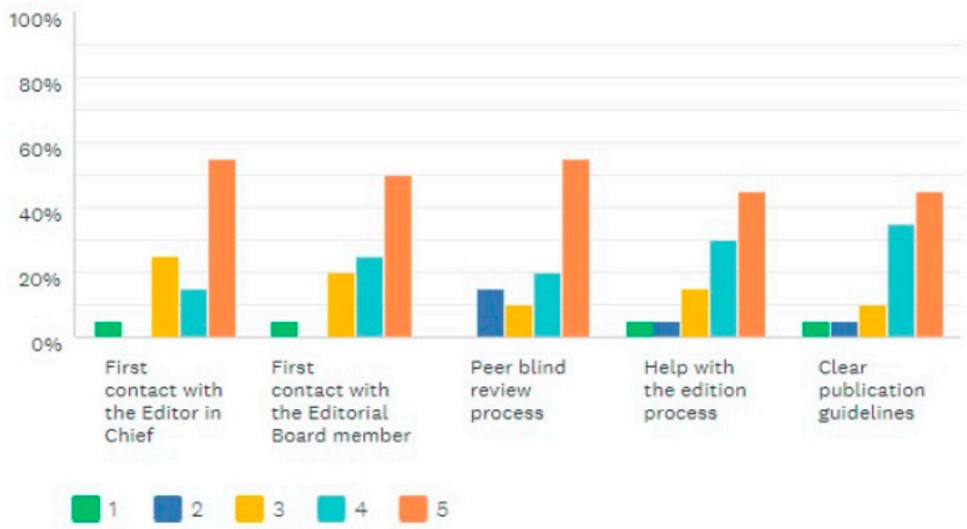

**Figure 6.** Q14. Please rate each component of the publication process for how much or little was helpful to you. Source: Prepared by the authors with data obtained from a survey (2021).

Figure 7 of our survey showed that 85% of the respondents said they increased their educational experience after publishing, which hints at emerging scholars pursuing their careers even further. Concerning their current occupation, 42% of the authors reported being employed full-time or part-time, while 36% of them reported being full-time or part-time students. Note that 64% of the authors are working in academia, while 6% are in the private sector, 15% work for the government sector, and 6% work for NGOs, respectively. This means that 36% of authors who, despite being outside of academia, seek to contribute to the generation of knowledge, thus an open access journal seems to be a good option. Albeit low proficiency in English can marginalize scholars, the editorial role of IAPSS can improve a manuscript's readability and improve scholars' overall writing skills (see Supplementary Materials). However, from the participants of this survey, a small number of authors have an intermediate proficiency of English. This is contained in the question on the rate of English proficiency, representing a more significant number of participants that have advanced in English proficiency or native speakers.

Figure 8 shows the responses to the question of whether they improved in their career. Of those surveyed, 58% reported that their career as emerging scholars improved, while 33% said their career was somehow enhanced. Out of the authors who answered that their employment had improved, 44% said that they grew to be more experienced in terms of the pub lication process, 41% said that it was vital for them to obtain a new item on their curriculum vitae, and 9% said they earned a unique position because of the number of their publica tions.

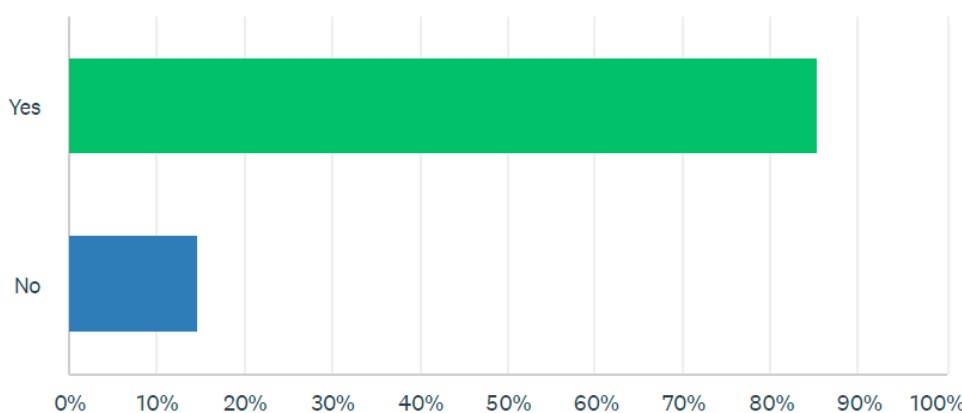

**Figure 7.** Q9. Did you increase your educational background after publishing with IAPSS *Politikon.* Source: Prepared by the authors with data obtained from a survey (2021).

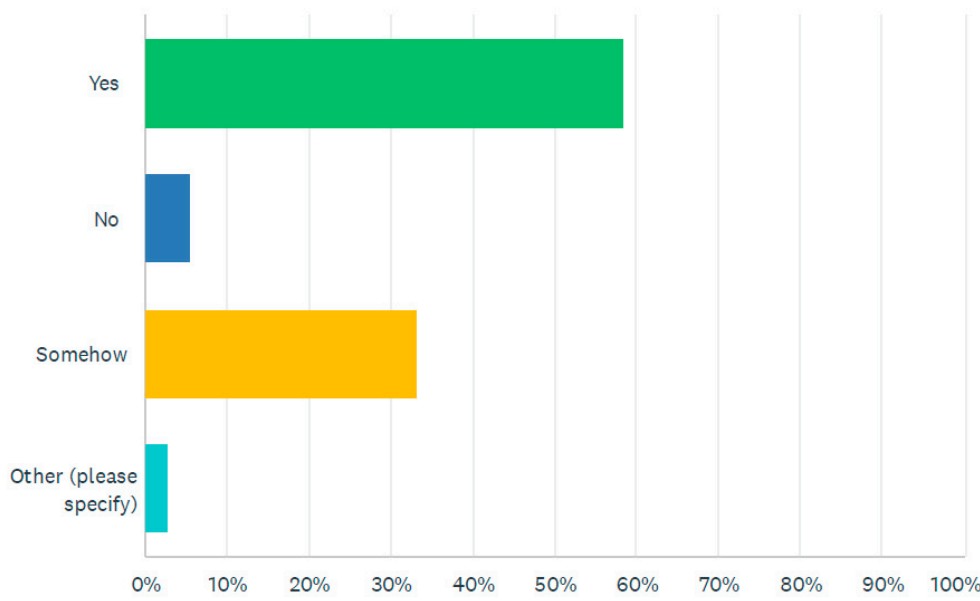

**Figure 8.** Q7. Early career improvement after publishing with IAPSS *Politikon.* Source: Prepared by the authors with data obtained from a survey (2021).

However, 14% said their educational background did not improve after pub lishing with *Politikon.* The evidence of this result was prevalent, as 47% of our find ings reveal that they obtained a higher position after publishing with *Politikon* (see Supplementary Materials). Even though we cannot discuss correlation, it could be a good indicator for emerging scholars and their research improvement. Additionally, this reminds us that researching and publishing have served as a condition for employment, promotion, and even maintaining one's job as the publishing choice of emerging scholars may produce a dramatic maximized career outcome and increase career success [11].

As Figure 9 shows, 65% of the surveyed scholars said they felt that their writing, reasoning, and research skills improved after publishing with IAPSS *Politikon*, which could indicate that the reviewing process was successful in encouraging them and helping them develop or enhance skills that are very important at the time of submitting a manu script to a journal. The process of publishing a paper in a journal is important in all its phases, because it is designed to help authors write a better paper through an improved research scheme that involves skills that can, in fact, be developed. The guidance provided by the editors and the whole team of the journal is vital for the improvement of the research skills

of the authors. Additionally, after this, the authors might feel engaged with the journals and their associations.

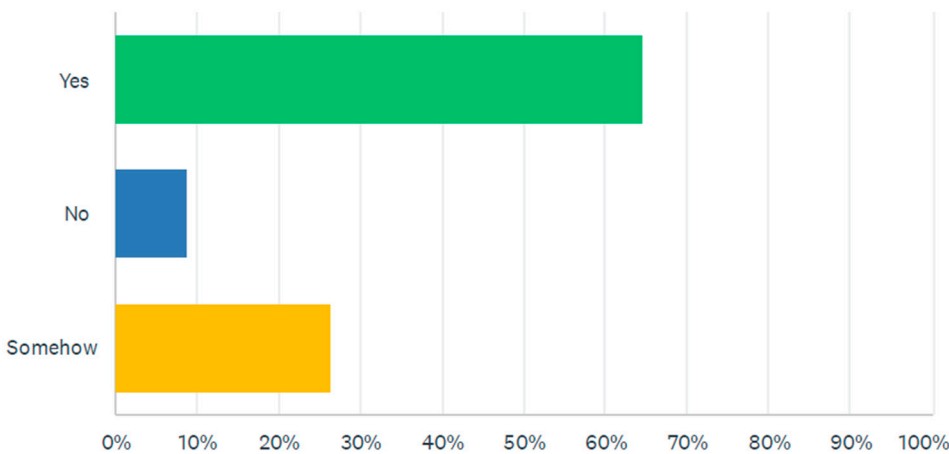

**Figure 9.** Q11. Do you feel that your writing/researching/reasoning skills have improved due to publishing with IAPSS *Politikon*. Source: Prepared by the authors with data obtained from a survey (2021).

For example, about 20% of them said their engagement with the association was complete after publishing, meaning that they would participate in world congresses or volunteer for other association activities. On the other hand, 5% of them said their engagement with the association is partial, which means they participate in the world congresses but do not participate in other association activities. Finally, less than 64%of them said their engagement with the association was light, which means that they only follow the association's activities through online platforms (see Supplementary Materials question 12).

It is also interesting to note the membership status of the authors who have published with IAPSS *Politikon*. Critical, 44 per cent of the respondents were already members of the association before publishing with the journal, which could mean that being part of the association somehow incentivizes them to publish with the journal.

In Figure 10, the responses indicated that 68% of the authors reported being advanced in English proficiency, while 24% reported being native English speakers. This might indicate also indicate that scholars around the world are in need of having English as an acquired language in order to publish with a famous journal. IAPSS has also understood this and created the journal Encuentro Latinoamericano, which is focused on Latin American scholars. However, it is important for any journal to take into consideration that these numbers might be similar, and that most of the people who are trying to publish articles are not English speakers and probably do not have enough resources to pay for the necessary assistance. Therefore, they might need special assistance, and the journal should be accommodating enough to provide for this. As discussed above, researching and publishing have remained problematic and complex for emerging scholars because of the problem of research funding [21] and the deficiencies of research and writing skills [22]. Other challenges to having a competitive career are language barriers, pressure to network, and career uncertainty [23].

In summary, the survey was conducted over six months in 2021, collecting data across multiple respondents with various backgrounds. Notably, 54% of respondents were male, 36% were female, and less than 10% preferred not to disclose their gender. Unfortunately, this illustrates that there are more men in political science and international relations than women. Therefore, much work still needs to be done to close the gender gap in academia and the aforementioned fields of knowledge. Moreover, authors come from 20 different countries and four continents: Europe with 42.4%, the Americas with 27.3%, Africa with 18.2%, and Asia with 12.1%.

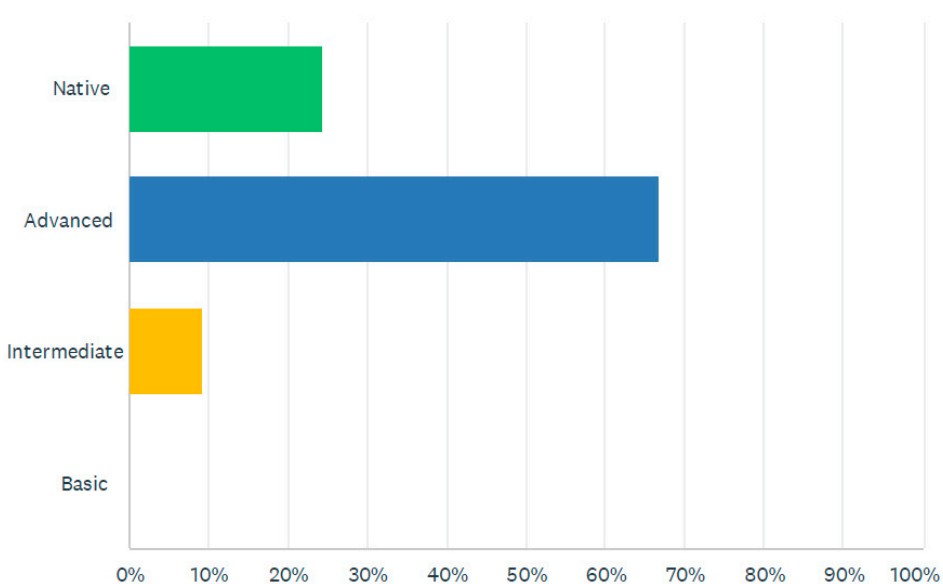

**Figure 10.** Q26. How would you rate your English proficiency. Source: Prepared by the authors with data obtained from a survey (2021).

Noticeably, 64% of respondents are from the Global South, ranging from Africa and Latin America to Asia and Eastern Europe. Although 9% did not indicate their country of origin, only 27% of respondents are from the Global North, ranging from North America to northern Europe. Interestingly, 52% of all the respondents in this study are under the age of 30. Notably, about 42% believed that the prestige of the IAPSS journal as well as its high-quality result and impact, guided their choice of journal, while some respondents, about 39%, identified the prestige of the student's association (IAPSS).

These results show the rating of components or phases of the publication process with the journal and how helpful that part was for them. The results indicate high satisfaction with the publication process, with the first contact with the editor-in-chief and the editorial board members, and the peer-review process scoring slightly better than the subsequent editing process and the clarity of the publication guidelines. The survey showed that 85% said they increased their educational experience after publishing. These results show that 64% of the authors are working in academia, while 6% are in the private sector, 15%work for the government sector, and 6% work for NGOs, respectively.

Lastly, it is noteworthy that 65% of the surveyed scholars said they felt that their writing, reasoning, and researching skills improved after publishing with IAPSS *Politikon*, which could indicate that the reviewing process was good enough to encourage them and help them develop or enhance skills that are very important at the time of sub- mitting a manuscript to a journal.

## 5. Final Considerations

This article examined the impact of IAPSS *Politikon* on emerging scholars' involvement in academic publishing. Using the results of a standardized and self-administered web survey as supporting evidence, we have argued that a journal such as *Politikon* helps reduce some of the tremendous pressures faced by emerging scholars worldwide that are identified in the literature. We have shown that publishing with *Politikon* has helped young researchers obtain more experience concerning the publication process and im- prove their research, writing, and analytical skills.

Additionally, most respondents said they increased their educational background after publishing, which might hint at these emerging scholars pursuing their careers further and further. Moreover, 50% of these scholars said they obtained a higher position after publishing with *Politikon*. We cannot discuss causality, but it could be a good indicator for early career researchers. Publishing with a reputable journal gives a scholar a higher

probability of improving their career. As mentioned before, for emerging scholars, the impact of researching and publishing is at the heart of career prospects to attain career goals [13] and to sustain a position in a career [14] that has been driven partly by institutional expectations and personal career ambitions [15]. Noticeably, IAPSS provides a publication platform that does more than help emerging scholars publish; it fosters a desirable skill set by prospective employers and also empowers the contribution of knowledge to the study of political sci ence and international relations while providing an alternative to Western and Eurocen tric scholarship. Although categorized within Global IR, this phenomenon arguably brings marginalized sources of knowledge systems to the forefront, and empowers the disadvantaged emerging scholars of the Global South. Furthermore, within the call for decolonization, this empowerment of the voice of marginalized Global South scholars enables not only vast knowledge systems and competing views of politics in action, but also indigenous sources of knowledge to be scribed in their languages (and their primary context).

However, the study has a few limitations, particularly data generation. While we relied on a limited sample of 33 responses, the survey encountered difficulties in obtaining feedback. The delay with repeated emails to respondents resulted in the second round of surveys that increased the number of data generation. The response, however, was meaningful due to the success of the target population. For our future research, a couple of avenues can be mentioned. First, further research should target more respondents across the world to identify the overall impacts of *Politikon* on non-academics in different professions. Second, the way in which *Politikon* contributes should be studied through a mixture of interdisciplinary publications, to a series of issues in public administration, budgetary allocation, and financing other than social sciences.

Finally, to other journals, we recommend trying to reach out to young scholars around the world, given that their contributions to the academia are of a high value and they can also benefit from publishing in a high quality journal. This would make the journals more inclusive and diverse.

**Supplementary Materials:** The survey administered to the authors who published with IAPSS *Politikon* is available at https://www.mdpi.com/article/10.3390/publications11010012/s1. The figures are part of this paper and the rest of the data have not been published elsewhere.

**Author Contributions:** Conceptualization, N.C. and L.N.; Methodology, A.M.F.; Formal Analysis, A.M.F. and L.N.; Investigation, A.M.F., N.C. and L.N.; Data Curation, A.M.F.; Writing—Original Draft Preparation, A.M.F., N.C. and L.N.; Writing—Review and Editing, A.M.F., N.C. and L.N.; Visualization, A.M.F.; Supervision, A.M.F., N.C. and L.N.; Project Administration, A.M.F., N.C. and L.N. All authors have read and agreed to the published version of the manuscript.

**Funding:** This paper has not had any funding by any institution or university. It has been done with the resources of the three authors involved in the project.

**Data Availability Statement:** The data presented in this study are available on request from the corresponding author. The data are not publicly available due to the fact that it contains private details from the respondents. The data presented in this study are available in the survey that can be found at the following link https://www.surveymonkey.com/r/MGZTFL2 (accessed on 24 November 2022). If you require access to the data, please provide an email address so that we can proceed with the access.

**Acknowledgments:** The authors would like to acknowledge IAPSS for their support in helping us contact the authors, so that we could get in touch with them and explain the idea of the project and ask for their collaboration with the survey.

**Conflicts of Interest:** The authors declare no conflict of interest.

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
