# Peer review of "Emerging Scholars in Academia: An Analysis of the Impact of IAPSS Politikon in the Academic Careers of Its Authors"

_publications, doi:10.3390/publications11010012_

Round 1

Reviewer 1 Report

The manuscript presents a compelling case about the disparities present in publication dynamics in journals belonging to the Global South. Its literature review is adequate. The data presented is small but could be satisfactory. Unfortunately, the authors need to revise the methodology (detail the case study methodology and employ data/ citations to back up their sampling decision, as well as present their rationale in a clearer manner) and the presentation of results.  

This proposal lacks a discussion that triangulates the literature review and the findings. After that, the conclusions need address the questions they themselves raised while writing the literature review. I’ve attached a word file with comments on most pages (P# = Page Number) in hopes that the manuscript can be polished up to standard with the journal requirements.

Reviewer 2 Report

First of all, I think the subject is important: how can open access journals support early career researchers. Thus, I would like to make sure this article is published.

However, there are several issues that need attention in my opinion.

Section 1:

“However, most open access journals are predatory journals or “scammers” that publish articles or unscientific papers for financial benefits (Beall, 2012).”
This statement is not true! While the problem of predatory publishers exists, the work of Beall is not the best source on this. A short discussion the criticism can be found on the Wikipedia page (https://en.wikipedia.org/wiki/Beall%27s_List) – see section “Criticism”. A better source might be the book by Linacre, S. (2022). The Predator Effect: Understanding the Past, Present and Future of Deceptive Academic Journals. Against the Grain (Media), LLC. https://doi.org/10.3998/mpub.12739277. This needs to be addressed throughout the article!

“Since its emergence in 2001, Politikon, a leading academic peer-reviewed journal, has had 54 volumes of scholarly articles published four times a year and distributed online free.” This sentence returns a bit further in the same section. Please remove one of those sentences.

Section 2:

I would suggest to create two subsections, to make it easier for readers:

·         Early career researchers

·         Peer review and predatory journals

Section 3:

An important omission: how many requests have been sent, and what is the response rate of the survey? It is now impossible to assess whether 33 response are a good or a less positive result.

Furthermore: the text mentions 36 results, while the figures lists 33 results. Please make sure the correct number of responses are reported!

Section 4:

“The survey was conducted from June 10 to December 31, 2021, collecting 36 answers. It is worth saying that the data were collected in two waves. The first one from June 10 to June 18, and the second one from December 1 to December 31st. All respondents confirmed that they have published at least one article with Politikon.” This should be part of section 3.

Figure 2: I would suggest to order the countries based on the values. This makes it easier to see where the largest number of respondents resides, while still displaying the global diversity

“It is worth saying that Politikon is indexed in the Directory of Open Access Journals (DOAJ), ERIH PLUS, International Political Science Abstracts, and J-Gate (see figure 5). A way of interpreting this is that it seems that even those for whom certain indexing services are essential do sometimes go for journals that are not indexed, indicating that other reasons affect the choice of the journal as well.” How do you know that the authors know about this?

“Albeit low proficiency in English can marginalize scholars (see figure 10), the editorial role of IAPSS can improve a manuscript’s readability and improve scholars’ overall writing skills (see Annex).” The figure shows that a large majority of the respondents report being proficient in English. This seems to be a contradiction. Could you explain this?

“In summary, […] after publishing.” This text has already appeared earlier. Please rewrite and make it shorter.

Last section: this section must be renumbered to section 5.

Overall:

The citations are not formatted in a consistent manner:

·         (Majumder, 2015). Comma before the year

·         (Osterloh & Frey 2015). No comma before the year

·         (Wilkins, Hazzam, Lean (2021). One bracket before the year

·         Etc.

I would strongly suggest to format all citations in a consistent manner. Zotero might be a helpful tool, that is freely available.

Round 2

Reviewer 2 Report

The revisions of the manuscript are - in my opinion - are a significant improvement.